# Research on Wave Energy Harvesting Technology of Annular Triboelectric Nanogenerator Based on Multi-Electrode Structure

**DOI:** 10.3390/mi13101619

**Published:** 2022-09-27

**Authors:** Chun Jie Wang, Fan Meng, Qiang Fu, Chen Hui Fan, Lin Cui

**Affiliations:** 1School of Electrical Engineering and Automation, Tianjin University of Technology, Tianjin 300384, China; 2Tianjin Complex System Control Theory and Application Key Laboratory, Tianjin 300384, China; 3School of Electrical and Electronic Engineering, North China Electric Power University, Beijing 102206, China; 4National Ocean Technology Center, Tianjin 300112, China

**Keywords:** annular triboelectric nanogenerator, energy harvesters, self-powered sensors, multi-directional efficient harvesting, output performance optimization

## Abstract

Triboelectric nanogenerators can convert wave energy into the electrical energy required by ocean sensors, but the problem of the low electrical output performance of triboelectric nanogenerators has always been a concern. In this paper, an annular triboelectric nanogenerator (A-TENG) composed of an annular outer shell and an inner ball is proposed to improve the electrical output performance of the triboelectric nanogenerator by optimizing the structural parameters and wave parameters. Using the control variables, the effects of structural parameters (structure size, number of electrodes, electrode spacing, inner ball diameter, and number of inner balls) and wave parameters (wave frequency and wave amplitude) on the electrical output performance of the A-TENG were studied by combining COMSOL simulation and experimental research. The experimental results show that increasing the diameter and number of inner spheres can improve the open-circuit voltage between electrodes; the multi-electrode structure can improve the electron transfer rate and efficiently collect wave energy in all directions; and within the range of fixed sea conditions, there is an optimal annular size, which has the advantages of good electrical output performance and small size. The electrical output performance of the A-TENG can be greatly improved by optimizing the structural parameters. There are optimal wave parameters, such that the A-TENG can maximize the ocean wave energy conversion. This low-cost, long-life, efficient, and reliable energy harvesting system is ideal for powering ocean sensors.

## 1. Introduction

In the era of rapid development of the Internet of Things, the demand for sensors is increasing. Therefore, research on sustainable, long-lasting energy harvesters for powering sensors is increasingly important [1]. Energy harvesting technology is also being widely studied, in most cases, the energy harvester mainly uses one of the environmental energy sources to generate electricity, such as mechanical energy, i.e., body movement, machine vibration, wind, water waves, etc. [2,3,4,5,6]. Because different environments have different energy sources available for energy harvesting, energy harvesters suitable for each energy source must be developed [7]. The intelligent construction of marine information requires unmanned aerial vehicles, such as buoys, unmanned boats, and underwater gliders, and is also inseparable from a large number of sensors as monitoring methods. At present, most of the power supply methods of ocean sensors are battery power supply, solar cell power supply, and wire-connected power supply, which have shortcomings, such as short life, large volume, high maintenance cost, and harm to the environment, and cannot meet the continuous supply demand [8,9,10,11]. Therefore, the power supply problem of ocean sensors has always been a concern. The ocean is a huge treasure house of energy, covering about 70 percent of Earth’s surface. The energy obtained from the ocean is called blue energy, which is inexhaustible and convenient [12], and the wave energy in it has not been well developed or utilized. Wave energy has the advantages of wide distribution, high energy density, renewability, and pollution-free. Converting wave energy into electrical energy can be a good power supply for ocean sensors. As a traditional method of wave energy conversion and transmission, electromagnetic generators need a stable and high operating frequency (50–60 Hz) to obtain an effective output [13,14] and have the disadvantages of high cost and complex structure. Most of the domestic oceans are quiet sea areas, and wave energy has the characteristics of low frequency, irregularity, low amplitude, and random direction of flow. Therefore, electromagnetic generators have obvious limitations in the acquisition of low-frequency waves [15,16,17].

In 2012, academician Wang Zhonglin proposed a new way of generating electricity, triboelectric nanogenerators (TENGs). A TENG is a new type of energy conversion device that converts mechanical energy into electrical energy based on the principle of combining triboelectric electrification and electrostatic induction [18]. TENGs use Maxwell’s displacement current as the driving force [1,19] and are the fourth recognized high-efficiency environmental mechanical energy harvesting power generation technology after electromagnetic induction [20], piezoelectric [21] generators, and electrostatic [22] generators. In 2019, Li et al. [23] conducted a more detailed quantitative analysis of the four working modes of TENGs by establishing a theoretical analysis model and verified the consistency of the results of the analysis models with finite element analysis, providing a theoretical basis for the structural design. TENGs show great advantages in low-frequency wave energy harvesting with the advantages of light weight, low cost, simple structure, clean, and pollution-free [24,25,26]. The output power density of a TENG is proportional to the square of its triboelectric surface charge density since the open-circuit voltage and short-circuit current are proportional to it at the same time [27,28]. In 2022, Zheng et al. [29]. improved the electrical output performance by studying the quantitative energy conversion analysis of triboelectric nanogenerators and also promoted the development of energy conversion efficiency theory. The theoretical energy conversion efficiency of TENG is as high as 85 percent, while in practical applications, the actual output performance of TENGs is much lower than this value. Therefore, the improvement of the electrical output performance of TENGs has always been a concern.

In this paper, a triboelectric nanogenerator with an annular multi-electrode structure is proposed. The working mode adopts the independent layer mode, the independent friction layer adopts nylon (nylon) balls, the other friction layer adopts polytetrafluoroethylene (PTFE) film, and the electrode adopts copper (Cu) foil tape. In order to improve the electrical output performance of the A-TENG, the effects of its structure size, electrode number and spacing, inner ball diameter and amount on the electrical output performance of triboelectric nanogenerators were comprehensively studied. In addition, the influence of different wave frequencies and amplitudes on the electrical output performance of the triboelectric nanogenerator was also studied. For random and irregular waves and harsh marine environmental conditions, annular triboelectric nanogenerators may be one of the most suitable triboelectric nanogenerators for harvesting wave energy.

## 2. Theoretical Model and Working Principle

In this paper, an annular triboelectric nanogenerator (A-TENG) is designed that can efficiently collect low-frequency wave energy in all directions. The structure of the A-TENG is shown in Figure 1a. Here, we can see the arrangement of the copper foil tape, the copper foil tape with polytetrafluoroethylene (PTFE) film attached to the surface, and the nylon balls placed inside. When moving under the action of waves, the nylon balls inside the annulus roll back and forth between the electrodes inside the annular to provide an alternating current for the external load. The rolling design greatly reduces the energy loss caused by friction and improves energy conversion efficiency. In order to improve the electrical output performance of TENGs, a multi-electrode structure is adopted so that the movement period of the nylon ball inside the annulus is short, and the charge-transfer frequency is high, which increases the number of cycles of energy conversion. The short-circuit current is significantly improved, and the charge transfer frequency is improved. The annular structure can achieve high-efficiency acquisition of wave energy in any direction. The outer diameter of the annulus is 70 mm, the inner diameter is 35 mm, and the diameter of the nylon ball is 28 mm. Figure 1b shows the cross-sectional view of the fabricated A-TENG.

The working principle is based on the triboelectric effect [19]. When the ball rolls close to the inner wall of the annular shell, it causes a constant change in the space potential, and a potential difference is generated between the different copper electrodes so that an alternating current signal is generated in the wires connecting the different copper electrodes, and conversion of the mechanical energy to electrical energy is realized. The working principle of TENGs is shown in Figure 1c. nylon balls and PTFE film are used as dielectric friction materials, and copper foil tape is used as a sensing electrode. Under the action of the wave force, the nylon ball starts to roll freely inside the annulus. After multiple cycles of rubbing with PTFE film, the nylon ball becomes positively charged, and the PTFE film is charged with the same negative charge. In the initial state, nylon balls are located above Electrode 1. Due to electrostatic induction, electrons are transferred from Electrode 2 to Electrode 1. Electrode 1 is negatively charged to shield the positive charge of nylon, and the other electrodes are positively charged to shield the negative charge of PTFE. When the ball rolls from Electrode 1 to Electrode 2, due to electrostatic induction, negative charges flow from Electrode 1 to Electrode 2 through an external load, thereby generating a current; when the ball rolls to the position as shown by Electrode 2, all negative charges from Electrode 1 are transferred to Electrode 2. When the ball moves from Electrode 2 to Electrode 3, the charges flow in the opposite direction, generating a current of opposite polarity. Therefore, a current peak appears every two adjacent electrodes. As the nylon ball moves backward, back and forth between Electrode 1 and Electrode 2, the current also flows. An external load is connected between two adjacent electrodes, and the generation of the potential difference drives electrons to flow between the electrodes. Based on this fundamental relationship, the A-TENG can generate a large open-circuit voltage due to the relatively small inherent capacitance *C*.
(1)$$Voc=\frac{Qsc}{C}$$

The short-circuit current Isc not only depends on the TENG’s intrinsic parameter $\frac{dQsc}{dx}$ but also on the external motion speed *v* as seen below:(2)$$Isc=\frac{dQsc}{dt}=\frac{dQsc}{dx}\frac{dx}{dt}=\frac{dQsc}{dx}v$$

In the above equation, Voc means the open-circuit voltage, Qsc means the short-circuit charge, *C* means the capacitance, and Isc means the short-circuit current.

Figure 1d shows the potential distributions between two adjacent electrodes in different states on a two-dimensional plane. It can be seen that the change in the spatial position of the nylon charged ball relative to the torus shell results in a time-varying spatial potential distribution, which causes the potential difference between the two adjacent Electrodes 1 and 2 in the torus shell. This drives current to flow in the external circuit.

## 3. Electrical Output Performance of a Single Triboelectric Nanogenerator

### 3.1. Influence of Structural Parameters on Electrical Output Performance

Considering the effect of the structural parameters of the A-TENG on the electrical output performance. According to the different sizes of the annulus when it is moved by the wave force, the displacement and the swing angle are different, and the influence on the electrical output performance of the triboelectric nanogenerator is also different. The influence of the swing angles of annuli of different sizes under the action of waves on the electrical output performance was studied under fixed sea conditions. Using COMSOL software to establish a two-dimensional model for fluid–structure interaction simulation analysis, the displacement of the triboelectric nanogenerator under the action of a regular sine wave was simulated. The displacements corresponding to the annular triboelectric nanogenerators at different wave frequencies and flow rates were analyzed [30,31]. The motion of a rigid body under waves can be decomposed into six degrees of freedom, namely, surge, sway, heave, roll, pitch, and yaw. This simulation starts from one of the degrees of freedom: roll. The coordinate system of the A-TENG is shown in Figure 2a. The displacement of the annulus is positive with the positive direction of the z-axis; the positive direction of the x-axis is the forward direction of the wave, R is the outer radius of the annulus, and r is the inner diameter of the annulus. Figure 2b shows the plane coordinate system of the A-TENG. Take point A on the annulus for research. When the annulus floats horizontally on the wave surface, the line connecting point A to the fixed point O coincides with the x-axis of the horizontal plane. When the annulus is affected by the wave and starts to move, the displacement of point A in both the x- and y-directions changes, and then the line connecting point A to the fixed point O forms an angle with the horizontal x-axis, that is, the swing angle of the annulus. The displacement in the x- and y-directions can be calculated using the tangent function. In this paper, there are four independent variables: horizontal flow velocity, vertical flow velocity, frequency, and annular size, and orthogonal tables are designed for the four levels of these four independent variables. The swing angle influences a dimensional annular triboelectric nanogenerator. Since A-TENG will be applied to harvest wave energy in China’s Bohai Sea, the independent variable parameters are set according to the Bohai sea conditions [32]. The specific parameter values are shown in Table 1 below.

Table 2 shows the results of the orthogonal multivariate analysis of the variance. As can be seen from Table 2, the significance of the horizontal flow velocity Vx is 0.036, the significance of the vertical flow velocity A is 0.157, the significance of the frequency f is 0.04, and the significance of the size S is 0.038. As specified in orthogonal ANOVA, a significance of less than 0.05 indicates that we are 0.995 confident in the result. It is concluded that the four variables have a significant influence on the swing angle of the annulus. Figure 2c is a contour plot of the estimated marginal mean values of annular swing angles corresponding to the four independent variables at different levels. The four line graphs show the influence trend of each independent variable on the swing angle of the annulus. It can be seen from the graph that, with the increase in the horizontal flow velocity Vx, the estimated marginal average value of the swing angle increases and reaches the maximum at Vx4; as the vertical flow velocity Vy increases, the estimated marginal mean value of the swing angle increases and reaches the maximum at Vy4; with the increase in the frequency f, the estimated marginal mean value of the swing angle gradually decreases and reaches the maximum at f1; as the annular size S increases, the estimated marginal mean of the swing angle changes continuously and reaches a maximum at S3. An optimal combination of Vx4Vy4f1S3, which maximizes the swing angle of the A-TENG, can be obtained in which the horizontal flow velocity Vx is 0.4 m/s, the vertical flow velocity A is 0.4 m/s, the frequency f is 0.5 Hz, and the size is S3, i.e., the outer radius of the annulus is 70 mm, and the inner diameter is 35 mm. Figure 2d shows the error bar of the swing angle. It can also be seen more clearly from the figure that the mean value of the swing angle is the largest in the 16th group, which corresponds to the best combination of the four independent variables of Vx4Vy4f1S3, which includes a horizontal flow velocity Vx of 0.4 m/s, a vertical flow velocity A of 0.4 m/s, a frequency f of 0.5 Hz, and size of S3, i.e., the outer radius of the annulus is 70 mm, and the inner diameter is 35 mm.

It is concluded that in this fixed sea state, the annular swing angle reaches the maximum value when the size is S3. As the swing angle of the annular triboelectric nanogenerator increases, the centrifugal force and rolling speed of the nylon balls inside the annulus improve, which increases the contact and friction between the nylon balls and the PTFE film. With the enhancement of contact and friction between the nylon balls and the PTFE film, the triboelectric effect enhances, the frictional contact area increases, and the electrical output performance improves. This simulation determined that there is an optimal annular size within the range of fixed sea conditions that maximizes the swing angle of the A-TENG when it is moved by wave force, thereby improving the electrical output performance of the triboelectric nanogenerator. Moreover, compared with the larger size A-TENG, this optimal annular size has the advantages of better electrical output performance and smaller size.

After determining the size of the annular structure, the effect of nylon balls of different diameters and quantities on the electrical output performance of the A-TENG was studied by using the control variables (changing the multi-factor problems into multiple single-factor problems and changing only one of the factors so as to study the influence of this factor on things, to study them separately, and, finally, to solve them comprehensively.) to observe the open-circuit voltage between two adjacent electrodes when nylon rolls over them. The diameters d of the nylon balls are selected as 1.4 cm, 2.1 cm, and 2.8 cm, as shown in Figure 3a. With the increase in the diameter of the ball, the open-circuit voltage gradually improves. As the number of balls n increases from one to three, the open-circuit voltage also improves, as shown in Figure 3b. This is because the increase in the diameter and quantity of small balls not only increases the frictional contact area between the nylon ball and the PTFE film but also improves the total amount of positive charges on the surface of the small ball, which increases the degree of influence on the space potential and improves the open-circuit voltage between two adjacent electrodes [33]. The diameter of the ball is close to the inner diameter of the annulus, which can constrain the trajectory of the ball so that the ball rolls along the inner wall of the annulus all the time, thereby reducing back and forth collisions with both sides of the inner wall of the annulus.

The voltage is output by the electrodes, so the arrangement position of the electrodes has an important influence on the voltage. Therefore, the effect of electrode spacing and quantity on the electrical output performance of the A-TENG was studied. When the rolling position of the nylon ball changes, the electrode voltage changes, so the middle position of each electrode is selected to study the open-circuit voltage between two adjacent electrodes. Taking the electrode width as the standard, the electrode spacing is selected to be 0.5, 1, and 1.5 times the electrode width. As shown in Figure 3c, the open-circuit voltage between two adjacent electrodes improves with an increase in the electrode spacing. The inner electrode of the annular adopts a grid electrode structure. The non-grid electrode type of TENG needs the displacement of the entire plate length to make the induced charge complete one movement, while the grid electrode type of TENG only needs a displacement of a unit length to completely transport the induced charge, which greatly improves the energy conversion efficiency. The displacement of the cell length, in turn, enables the recirculation of the induced charge. Therefore, in the unidirectional sliding process over the entire length of the TENG, the charge can be induced (2N−1) times in total, where N is the number of grid electrode units. For calculation of the contact area during the sliding process of the two surfaces inside the grid electrode TENG, the following equation is the total amount of induced charges generated during a single sliding process of the grid electrode TENG along the entire length: (3)$$Q=Nq^{\prime }+|-q^{\prime }\left(N-1\right)|+q^{\prime }\left(N-1\right)+\ldots +\left|-q^{\prime }\right|+q^{\prime }$$
(4)$$=Nq^{\prime }+2q^{\prime }\sum_{i=1}^{N-1}id\;$$
(5)$$=\frac{(2q^{\prime }N)N}{2}\;$$

The maximum total amount of the induced charges generated by the non-grid electrode TENG with the same length is $2q^{\prime }N$. The formula shows that the total amount of the short-circuit-induced charge improves linearly with the increase in grid electrode density.

Because the ball and the copper electrode carry different kinds of charges, the potential difference between the two adjacent Copper Electrodes 1 and 2 reaches the maximum value when the ball is located at a certain position on the surface of Copper Electrode 1 or 2. When the number of copper electrodes is small, it means that the area of each copper electrode is large. When the small ball is located at a certain position on the surface of Copper Electrode 1, it will only significantly change the local potential of the small ball on the surface of Copper Electrode 1. However, most of the other positions of Copper Electrode 1 are still far away from the ball, so the overall potential of Copper Electrode 1 is not greatly affected by the ball, so the potential difference between Copper Electrodes 1 and 2 is small; when the number of copper electrodes is large, it means that the area of each copper electrode is very small, so the positions of Copper Electrodes 1 and 2 are very close. When the small ball is located at a certain position on Copper Electrode 1, Copper Electrode 2 is also very close to the small ball, so the potential difference between 1 and 2 is also smaller. Therefore, if the area of each copper electrode is too large or too small, the potential difference between two adjacent electrodes reduces.

In the annular structure, the electrode area is proportional to the electrode width, and the electrode width is the same as the electrode spacing. Select the number of electrodes *N* = 4, 8, and 12, and observe the open-circuit voltage between two adjacent electrodes, as shown in Figure 3d. It was observed that with the increase in the number of electrodes, the open-circuit voltage gradually decreased. The open-circuit voltage was the largest with four electrodes, but the slope of the open-circuit voltage was small; when there are eight electrodes, the open-circuit voltage is relatively reduced. When there are twelve electrodes, the open-circuit voltage is the smallest. The slope of the open-circuit voltage between adjacent copper electrodes is large and it is the same when there are eight and twelve electrodes. The slope refers to the improvement of the open-circuit voltage. A relatively large slope can ensure better charge transfer efficiency and device performance of triboelectric nanogenerators, and there is such a relationship between every two adjacent copper electrodes. At the same time, the multi-electrode structure inside the annulus can make the movement direction of the inner nylon balls always consistent with the electrode direction and improve the capture efficiency of wave energy.

### 3.2. Influence of Wave Parameters on Electrical Output Performance

Controlled wave excitation in a single favorable direction is not possible in a typical irregular ocean due to constantly changing conditions and wave characteristics. The motion of the A-TENG under the action of wave force is shown in Figure 4a, which can be decomposed into six degrees of freedom, namely, heave, surge, sway, yaw, pitch, and roll. In the best case, all of the kinetic energy of the generating unit is absorbed. Axisymmetric geometry has often been studied before because it absorbs energy independent of wave direction. For axisymmetric oscillating bodies, since there is no restoring force in the three directions of heave, sway, and yaw, the degrees of freedom in these three directions need to be restricted. There are restoring forces in the three directions of roll that can force the oscillating body to return to the equilibrium position [34]. The A-TENG is an axisymmetric structure. In order to study the dynamic characteristics of the A-TENG, a wave simulation platform is designed and built in this paper. The physical map and the schematic diagram of the platform structure are shown in Figure 4b. The motion state of the A-TENG under the three degrees of freedom of heave, pitch, and roll can be simulated. The frequency range of the simulated wave is 0.5–2.5 Hz, and the simulated wave amplitude is 60–120 mm. A single A-TENG was fixed on a wave simulation platform to study the effects of wave frequency and amplitude on the electrical output performance of the A-TENG.

Figure 4c,d shows the open-circuit voltages generated by the motion of the A-TENG at different frequencies and different amplitudes under the heave degree of freedom. Using the control variables, the fixed amplitude is 120 mm, and it is found that the open-circuit voltage first improves and then reduces with the increase in frequency. Moreover, the maximum output is reached at a frequency of 1.5 Hz. When the frequency increases, the swing amplitude of the annulus remains unchanged by the wave motion, and the swing angular velocity increases, which improves the centrifugal force and rolling speed of the nylon ball, increasing the contact and friction with the PTFE film and the frictional electrification effect. The contact surface charge density increases, and the open-circuit voltage and the amount of transferred charge improve. The fixed frequency is 1.5 Hz. With the increase in wave amplitude, the open-circuit voltage gradually improves and reaches the maximum value when the amplitude is 120 mm.

### 3.3. Results and Disscussion

In this chapter, the electrical output performance of the A-TENG is verified and experimentally tested from two aspects of structural parameters and wave parameters through simulation and experiment. Firstly, the influence of structural parameters on the electrical output performance of the A-TENG was studied by simulation, including the diameter and number of nylon spheres, the distance and number of electrodes, and the size of the annulus. When the diameter and number of nylon beads increase, the total amount of positive charges on the nylon surface improves, and the nylon beads have a significant impact on the space potential, which improves the open-circuit voltage between electrodes; when the electrode spacing is too large or too small, the rolling of nylon balls has little effect on the space potential of adjacent electrodes and reduces the potential difference between adjacent electrodes. The multi-electrode structure can improve the electron transfer rate and efficiently collect wave energy in all directions. The optimal structural parameters were finally determined, namely, the diameter of nylon beads was 28 mm, the quantity was three, the number of electrodes was eight, the electrode spacing was the same as the electrode width, and the annular size was 70 mm in outer diameter and 35 mm in inner diameter. Second, the effects of wave parameters on the electrical output performance of the A-TENG, including wave frequency and amplitude, were tested using a wave simulation platform. In the range of fixed wave parameters, with the increase in frequency, the open-circuit voltage of the A-TENG improved first and then reduced; with the increase in amplitude, the open-circuit voltage of the A-TENG improved. When the wave frequency is 1.5 Hz, and the amplitude is 120 mm, the open-circuit voltage of the triboelectric nanogenerator reaches the maximum value, and the maximum output electric energy is reached.

## 4. Conclusions

In conclusion, this paper proposes a novel triboelectric nanogenerator with an annular-shaped multi-electrode structure that can effectively harvest a large amount of wave energy. This structural design is suitable for the application of wave energy collection and has the following four key advantages: First, the design of the annular structure can flexibly and efficiently collect wave energy in any direction of motion, including up and down, left and right swing, and rotation motions. It is not easy to overturn on the sea surface and work continuously and stably. Secondly, the inner electrodes of the annular are arranged in a multi-electrode structure, which increases the frequency with which the charged ball rolls over the electrodes, improves the charge transfer efficiency, and improves the electrical output performance. Third, the selection of the optimal size ensures that the annular-shaped multi-electrode triboelectric nanogenerator can collect more wave energy within a fixed sea state range, which greatly improves the electrical output performance and has the advantage of small size. Finally, when multiple TENGs are integrated into a network through a series or parallel (Network), this can effectively collect a large amount of blue energy from the ocean. The “blue energy dream” will be realized in the near future. In this study, the optimized design of the diameter and number of rolling balls, the electrode structure, and the annular size was completed, and the electrical output of the A-TENG in motion in three degrees of freedom was analyzed. The results show that this structural optimization design can effectively improve electrical output performance. The A-TENG achieves the best electrical output performance when the fixed wave frequency is 1.5 Hz and the amplitude is 120 mm. This research greatly improves the electrical output performance of the TENG, which is of great significance in developing a system that can collect a large amount of low-frequency wave energy, and has made significant contributions in the fields of marine monitoring and the power supply of marine micro-devices.

## Figures and Tables

**Figure 1 micromachines-13-01619-f001:**
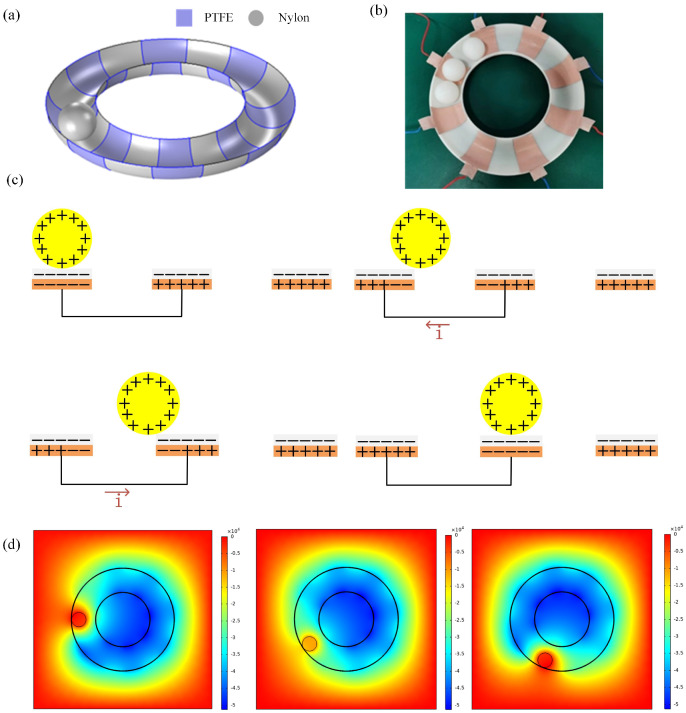
Structure, working principle, and potential distribution of the device based on an annular-shaped triboelectric nanogenerator (A-TENG). (**a**) Schematic diagram of the structure of the A-TENG. (**b**) Photo of the physical section of the fabricated A-TENG. (**c**) The working principle of the A-TENG in the working mode. (**d**) The potential distribution at different rolling angles.

**Figure 2 micromachines-13-01619-f002:**
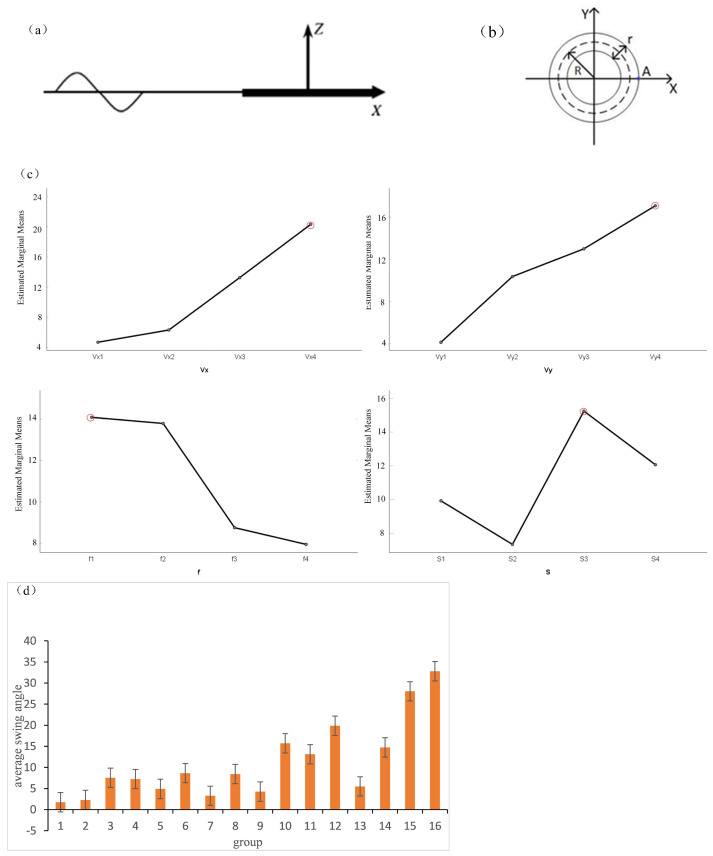
(**a**) The coordinate system of the A-TENG. (**b**) The plane coordinate system of the A-TENG. (**c**) The contour plot of the estimated marginal mean value of the annular swing angle corresponding to different levels of the four variables. (**d**) The error bar of the swing angle plot (standard error).

**Figure 3 micromachines-13-01619-f003:**
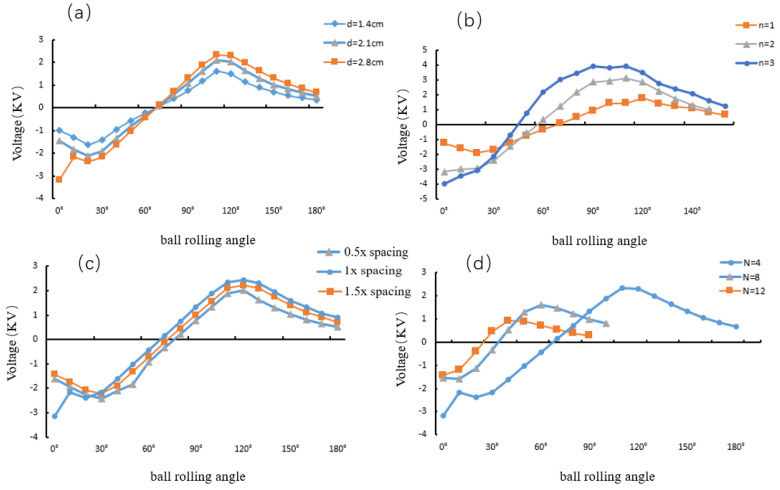
(**a**) Open-circuit voltage for different nylon ball diameters. (**b**) Open-circuit voltage for different quantities of nylon balls. (**c**) Open-circuit voltage for different electrode spacings. (**d**) Open-circuit voltage for different amounts of electrodes.

**Figure 4 micromachines-13-01619-f004:**
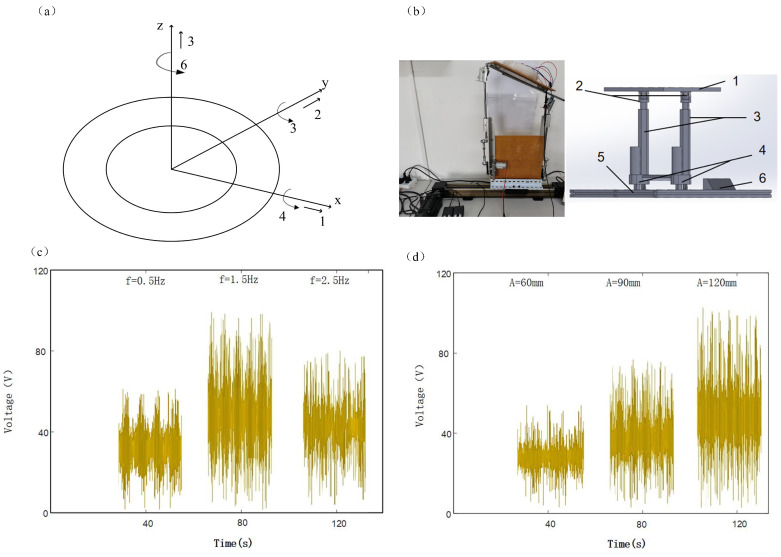
(**a**) Six degrees of freedom of the A-TENG under wave force (1. surge, 2. sway, 3. heave, 4. roll, 5. pitch, 6. yaw). (**b**) Schematic diagram of experimental platform and platform structure (1. glass platform, 2. connecting structure, 3. two electric push rods, 4. push rod base, 5. linear module slide table and slide rail, 6. master control platform). (**c**,**d**) Open-circuit voltage at different wave frequencies and different wave amplitudes under heave degrees of freedom.

**Table 1 micromachines-13-01619-t001:** The parameter values of the four independent variables.

Horizontal Velocity Vx (m/s)	Vertical Velocity Vy (m/s)	Frequency f (Hz)	Ring Size S (Major * Minor Radii/mm)
0.1	0.25	0.5	30 * 15
0.2	0.5	1	50 * 25
0.3	0.75	1.5	70 * 35
0.4	1	2	90 * 45

**Table 2 micromachines-13-01619-t002:** Results of orthogonal multivariate ANOVA.

Origin	Type III Sum of Squares	Degrees of Freedom	Mean Square	F	Salience
Corrected model	3.066 ${}^{a}$	12	0.255	10.086	0.041
Intercept	4.965	1	4.965	196.013	0.001
Vx	1.525	3	40.508	20.063	0.017
Vy	0.889	3	0.296	11.7	0.037
f	0.316	3	0.105	4.156	0.136
A	0.336	3	0.112	4.425	0.127
Error	0.076	3	0.025		
Total	8.107	16			
Corrected total	3.142	15			
a. R-square = 0.937 (Adjusted R-square = 0.686)					

## Data Availability

Not applicable.

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
