# Peer review of "Research on Wave Energy Harvesting Technology of Annular Triboelectric Nanogenerator Based on Multi-Electrode Structure"

_micromachines, 2022, doi:10.3390/mi13101619_

Round 1

Reviewer 1 Report

This work reported a triboelectric nanogenerator with an annular multi-electrode structure, which can stably and efficiently harvest wave energy in all directions. For random and irregular waves and harsh marine environmental conditions, annular triboelectric nanogenerators may be promising when harvesting wave energy. This work is interesting and well organized. It is suggested that this work may be accepted for publication in Micromachines after major revision by referring to the following comments.

1.      The paper should uniformly use fig or Fig.

2.      The figures with low definitions of the paper are not clear enough. Please use higher quality figures.

3.      The authors claim that this simulation study determines the optimal ring size within the range of fixed sea conditions, improves the electrical output performance of the triboelectric nanogenerator. The error bar or more comparative experiments should be added in Figure 2.

4.      In order to highlight the advantages of this work, some important works of TENG should be referred, e.g., Adv. Energy Mater. 2018, 8, 1800705 (DOI: 10.1002/aenm.201800705); J. Mater. Chem. A, 2019, 7, 19485 (DOI: 10.1039/c9ta06525c); Adv. Mater. 2022, 34, 2202238 (DOI: 10.1002/ adma.202202238); Nature 2017, 542, 159-160 (doi:10.1038/542159a). ACS Nano. 2021,15, 258-287 (doi: 10.1021/acsnano.0c09803); NPG Asia Mater 12, 6, (doi:10.1038/s41427-019-0176-0).

5.      In line 163 of the paper, the author claims that the increase of the diameter and number of the balls increases the surface charge density of the spheres. The authors should provide evidence to support this explanation. As far as I am concerned, when the increasing of the diameter and number of the balls, enlarging the friction contact area plays more important role than enhancing the surface charge density of the spheres.

Reviewer 2 Report

The authors have submitted an article promoting “wave energy harvesting technology of annular triboelectric nanogenerator based on multi-electrode structure”. Micromachines is the targeted journal.

The article is fair-written, and the reader understands a triboelectric generator architecture is modelled to harvest vibrational energy from waves. Even if such converters are perfectly justified in this context, it is not seen as a brand-new topic and the contribution to the field is not convincing enough.

The article might be submitted to another journal fitting the research work led better.

Reviewer 3 Report

Dear Authors,

I have reviewed the article thoroughly and highlighted some of my serious reservations in attached digital file (pdf). Some important points are here as well:

1. Must improve abstract and conclusion.

2. Literature review is limited to a particular group and recent developments are missed out.

3. Results and discussion section is cut down too short.

4. Validation/comparison part is totally missed out (even for literature), RECHECK.

5. The significance of the research is not made very clear.

Regards,

Round 2

Reviewer 1 Report

1 Some cited references should be corrected,e.g., the authors and journal name of ref.24 are wrong and the title should be added. Corrected as ‘Li, X.; Lau, T.H.; Dong, G.; Zi, Y. A Universal Method for Quantitative Analysis of Triboelectric Nanogenerators. J. Mater. Chem. A 2019, 7, 19485-19494.’ Meanwhile, another important research, which has been mentioned before, about the quantitative energy conversion analysis of triboelectric nanogenerators (Adv. Mater. 2022, 34, 2202238. DOI: 10.1002/ adma.202202238) should be added.

2 The sentence from lines 173 to 178 is quite long. Changing it into two sentences if possible. And in line 177, ‘the friction The contact area is increased’ should be corrected.

3 The sentence from lines 178 to 182 is not easy for readers to understand. And the Good in line 181 should be corrected as good. Check the same mistakes in other sentences of this manuscript, e.g., the Electrical in line 184.

Reviewer 2 Report

The authors have considered the various advice provided during the first review process. The efforts to provide a more consistent Introduction and conclusion give a manuscript a revisited interest and a potential impact in the field. The manuscript might be accepted for publication in Micromachines once the remaining typos are removed.

Reviewer 3 Report

I hope the mentioned changes in digital file are updated as well.
